# On the Dynamics of Inferential Behavior while Reading Expository and Narrative Texts

**DOI:** 10.3390/brainsci14050428

**Published:** 2024-04-26

**Authors:** Yongseok Yoo

**Affiliations:** School of Computer Science and Engineering, Soongsil University, Seoul 06978, Republic of Korea; yyoo@ssu.ac.kr; Tel.: +82-2-820-0678

**Keywords:** reading, reading disabilities, inference, think aloud, dynamics

## Abstract

Inference plays a key role in reading comprehension. This study examines changes in inferential behavior while reading different genres. The inferential behavior of 28 students with reading disabilities (RDs) and 44 students without RDs was quantified while they read expository and narrative texts. First, the average rates of inference attempts and correct inferences were measured during reading. Then, the same rates were measured separately during early and late reading to see if there was a change in inferential behavior. The results show that the change in inferential behavior depends on the genre. While reading the expository text, both groups showed no significant change in their inference making. In contrast, while reading the narrative text, both groups showed higher rates of inference attempts, and only the students without RD showed a significant increase in correct inferences. The implications of these findings for the design of more engaging and effective reading programs are discussed.

## 1. Introduction

### 1.1. Inference Making during Reading

Reading involves complex interactions between multiple cognitive processes, such as letter–sound correspondence, phonological memory, word recognition, sentence processing, and comprehension. A deficiency in any of the low-level processes can lead to reading disabilities (RDs) [1,2]. For example, deficits in lexicography and phonics can lead to difficulties with word identification and dyslexia [1,2]. Higher-level cognitive processes, such as reading comprehension, are less well understood [2].

Inference has been shown to play a key role in reading comprehension, serving as a critical component in the construction of meaning from text [3,4,5]. Inference allows readers to fill in gaps in explicit textual information, facilitating deeper understanding and integration of knowledge. Previous research has emphasized its importance not only for comprehending literal content but also for engaging with the text at a deeper level [4], allowing for the application of prior knowledge and the anticipation of subsequent narrative developments [5]. Carlson et al. [6] emphasized the importance of assessing reading comprehension by measuring the process of making inferences while reading. They emphasized the importance of examining the cognitive processes that struggling readers use while reading, such as literal comprehension and thinking about the implications of the text, and suggested think-aloud tasks as a useful assessment method.

To assess readers’ comprehension processes, think-aloud tasks are used. The think- aloud task was originally used as a tool to demonstrate thinking while performing a given task by asking for the subjects to say out loud what they are thinking while performing the task [7]. A common way to administer the think-aloud task in reading is to ask students to read one sentence at a time and say anything that comes to mind. Research using the think-aloud task has found that readers make a variety of inferences while reading [8,9]. For example, Laing and Kamhi [10] divided third graders by reading level to see if they differed on a think-aloud task. They found that average students made more explanatory inferences than struggling readers. Carlson et al. [6] administered a think-aloud task to struggling readers in Grades 3–5 and found that the struggling readers were more likely to simply repeat what they had just read, while the control group attempted to elaborate on what they had read.

Recent studies show that the genre of the text can influence the inferences made while reading. The difference in inferences between reading narrative and expository texts was investigated [11]. The results showed that students used inferential comprehension more often when reading narrative texts than expository texts, and the researchers argued that the comprehension of expository texts was more difficult. Students may have difficulty with expository texts because they require more background knowledge and skills to make connections to what they are reading [12,13]. In [14], the relationship between the characteristics of undergraduate students and their ability to comprehend expository texts was investigated. The results showed that readers’ background knowledge plays an important role in making inferences during reading. In [15], the effects of genre on the reading comprehension of elementary school students were investigated. The results showed that individual differences in world knowledge were more related to the comprehension of expository texts than narrative texts. However, there is still limited research on how elementary students with and without RDs differ in making inferences for their comprehension of different types of texts, such as narrative and expository texts.

### 1.2. Contributions of the Study

The results of previous studies are based on average inference behavior during reading, under the assumption that inference is a stationary process. Given the complex nature of inference, we suggest that inference behavior may change dynamically during reading. In this study, we investigate changes in the inferential behavior of students with and without RD during reading, focusing on how readers adapt their reading strategies in response to different genres. This investigation is based on the hypothesis that inferential behavior is not static but rather evolves dynamically with the reading context, reflecting a sophisticated interplay between the text and the reader’s adaptation. These interactions of students with and without RDs were quantified as they read texts of different genres (expository and narrative).

This study strategically targeted specific age groups with and without RDs. The third and fourth grades play a crucial role in a student’s education, representing a pivotal point in the development of essential comprehension skills that are critical for later learning [13]. The ability to understand the meaning of a text and make inferences is expected to be mastered before fourth grade, and comprehension difficulties during this period have been linked to later academic underachievement in all subjects [16]. Thus, understanding challenges in these grades is vital to establish a strong foundation for future academic success. Therefore, this study focused on the inferential behavior of students with and without RDs during reading at this stage.

Specifically, we used the standard think-aloud protocol used in previous studies of reading [6,7,8,9,10]. The novel contributions of this study include the focus on inference making and its change over time. The research questions for this study were as follows.

First, when reading expository and narrative texts, do students with and without RDs differ in the rate at which they make inferences and the accuracy of their inferences?

Second, when reading expository and narrative texts, do students with and without RDs show a change in their inference making during reading?

## 2. Materials and Methods

### 2.1. Participants Information

Seventy-two third- and fourth-grade students in 14 public elementary schools in South Korea participated in this study. Each school had approximately 500 students enrolled. Within each grade, classes were divided into four or five sections, with 20–24 students in each class. None of the participants received any financial or social assistance.

Table 1 shows participants information. Among the participants, 28 students were identified as having RD by screening followed by the standardized Reading Achievement and Reading Cognitive Process (RA-RCP) tests [17] (scoring below the 16th percentile). The median scores of the students with RDs and without RDs were the 10th and 73rd percentiles, respectively. These students with RDs had intact word recognition and communication skills. The remaining 44 students without RDs served as the control group.

### 2.2. Data Collection Using the Think-Aloud Protocol

Three stimulus texts were taken from teachers’ guides for reading courses. The first text consisted of three short sentences and was used for the training session to familiarize the participants with the experiment.

The other two texts for the main session were similar in readability and came from different genres (expository and narrative). The texts for the main session were developed as follows. First, the researchers reviewed the textbooks, recommended books, and reading comprehension instruments for Grades 3–4 and extracted six candidate passages. Next, the content validity was checked by a group of experts; two elementary school teachers, a professor of language education, and a professor of Korean language and literature were asked to check whether the difficulty level and length of the passages were appropriate and met the purpose of the test. Then, two of the six passages were selected according to the experts’ opinions. Changes were made to adjust the difficulty level of the sentences and to shorten the length. The experts reviewed the revised texts again, and an agreement was reached among all the experts. Consequently, the expository text contained 10 sentences with 92 words and the average, minimum, and maximum numbers of words per sentence were 9.2, 5, and 15, respectively. The number of words in each sentence did not differ significantly between the first five sentences and the remaining sentences (*t*-test, *p* = 0.68). The narrative text contained 11 sentences with 92 words, and the average, minimum, and maximum numbers of words per sentence were 8.4, 4, and 13, respectively. The number of words in each sentence did not differ significantly between the first five sentences and the remaining sentences (*t*-test, *p* = 0.20). Each text was printed on A4 paper, with one sentence per line.

The tests were administered individually by a trained instructor in a quiet classroom. One sentence at a time was presented by the instructor while the other sentences were covered. The participants were asked to verbalize their thoughts right after reading each sentence, and those responses were transcribed.

The instructor training was implemented over the span of one week prior to the beginning of the study. This training encompassed the objectives of the assessment, the administration procedures, and considerations to observe during the administration of the assessment.

### 2.3. Coding

The reader’s response to each sentence was coded as one of correct inference, incorrect inference, and no inference by three trained evaluators (professors who specialize in learning disabilities). If their classifications for a response differed, they discussed the matter until agreement was reached.

The definitions of the three inference types are as follows. First, correct inferences are inferences that are made accurately based on the reading. Correct inferences may be explanatory, predictive, or associative inferences [9]. Explanatory inferences involve elaboration of the target sentence by relating it to previous sentences or background. Predictive inference is when the reader guesses what would come next based on what came before. Associative inference is making new inferences based on background knowledge or experiences that come to mind as the reader reads a sentence. Second, incorrect inferences are defined as inference attempts made by readers that are out of context or do not make logical sense. Third, no inferences are responses where no inference was made. They can be simple repetitions of what has been read or nonsensical answers.

### 2.4. Data Analysis

The frequency of the three inference types (correct inference, incorrect inference, and no inference) made by each reader were calculated for each text.

To measure any change in the inferential behavior, the frequencies during early and late reading were calculated separately. For the expository text, the frequencies corresponding to the first five sentences were measured for the early behavior, and those corresponding to the remaining five sentences were measured for the late behavior. For the narrative text, the frequencies corresponding to the first five sentences were measured for the early behavior, and those corresponding to the remaining six sentences were measured for the late behavior.

Statistical significance was measured as follows. Some of the measured frequencies did not pass the normality test using the Kolmogorov–Smirnov test. This is probably due to the small sample size and the limited range of frequencies. Therefore, nonparametric models were used for hypothesis testing. To compare the difference in inference frequency between students with and without RDs, the Mann-Whiney U test was performed. To test the change between early and late reading time, the Wilcoxon signed rank sum test was performed. All statistical analyses ware performed using SciPy version 1.9.1 [18].

## 3. Results

### 3.1. Comparison of the Frequency of Inferences between Students with and without RDs

Table 2 shows the frequency of inferences between students with and without RDs for expository and narrative texts. For the expository text, participants with RDs made significantly fewer correct inferences (*p* < 0.001, *t*-test) and significantly more incorrect inferences (*p* < 0.05, *t*-test) than those without RDs. The frequency of no inferences was also significantly higher for students with RDs than those without RDs (*p* < 0.001, *t*-test). For the narrative text, students with RDs made significantly fewer correct inferences (*p* < 0.001, *t*-test) than those without RDs. The frequency of no inferences was also significantly higher for students with RDs than for those without RDs (*p* < 0.001, *t*-test). Thus, the students with RDs attempted to make inferences less often, and their inferences were not correct more often than those of the students without RDs.

### 3.2. The Change in Inferential Behavior during Reading

Table 3 shows the inference frequencies during early and late reading of the expository text. The inference frequencies did not change between early and late reading for the both groups (*p* > 0.05, *t*-test).

Table 4 shows the inference frequencies during early and late reading of the narrative text. The frequency of correct inference increased during the late reading only for the students without RDs (*p* < 0.001, *t*-test), not for the students with RDs (*p* > 0.05, *t*-test). The frequencies of no inferences decreased during the late reading for both students without RDs (*p* < 0.001, *t*-test) and students with RDs (*p* < 0.01, *t*-test).

## 4. Discussion

The inferential behavior of the readers for different genres of the text was studied. First, the frequencies of inference types were quantified for students with and without RDs for the entire reading time. For both expository and narrative texts, the students with RDs attempted to make inferences less often, and their inferences were not correct more often than those of the students without RD. Second, the change in the inferences between early and late reading time was measured. For the expository text, the frequency of making inferences did not change between early and late reading. In contrast, for the narrative text, both groups attempted to make inferences more frequently during late reading, resulting in more correct inferences only for the students without RDs. Implications of this genre-dependent inferential behavior are discussed below.

The frequency of inferences of both groups remained constant throughout the reading of expository texts. This could be explained on the basis of previous findings that background knowledge plays a crucial role in reading expository texts [12,13,14,15]. The level of background knowledge related to the given text would determine the overall inference attempts and correctness but would not change significantly during reading.

In contrast, both groups increased the frequency of inferences in the later stages of narrative reading, whereas the frequency of inferences remained constant during the reading of expository texts. This change during reading has more to do with cognitive processes during reading than with previously acquired knowledge. It has been reported that students with RDs lack higher-order cognitive skills, such as focusing on essential content, extending sentence content, and predicting future events [19]. Our findings suggest that the dynamic engagement of such higher-order cognitive skills is more encouraged when reading narrative texts than expository texts. Thus, initiating reading instruction with story books may be an effective strategy for fostering comprehension skills. This approach leverages the natural inclination toward increased engagement with stories as they progress, potentially facilitating the development of inferential reasoning.

A clinical implication of our findings is the potential to improve diagnostic accuracy. Current standardized diagnostic tests measure reading ability over a relatively long period of time, which may not capture more subtle differences and changes in reading behavior during reading. Our findings suggest that monitoring changes in inferential behavior may provide additional information about reading behavior. This could improve diagnostic criteria and help differentiate between different types of reading disorders, allowing for more targeted interventions.

However, there remain several limitations that need to be addressed in future studies. In this study, representative expository and narrative texts were carefully selected by a group of experts. Investigating the generalizability of our findings to diverse texts is an important direction for future studies. In addition, another direction would be to confirm the dynamic change in inference behavior using other methods, such as neuroimaging technologies, and to understand the neural underpinnings of this change.

## 5. Conclusions

In this study, we investigated changes in inferential behavior while they read expository and narrative texts. Students with RDs participants showed lower rate of inference attempts, and these inferences were not correct more often than those of students without RDs. While reading the expository text, students both with and without RDs showed no significant change in their inference making. In contrast, while reading the narrative text, both groups showed higher rates of inference attempts, and only the students without RDs showed a significant increase in correct inferences. These findings suggest that consideration of genre is important in designing more engaging and effective reading programs.

## Figures and Tables

**Table 1 brainsci-14-00428-t001:** Participants information.

Grade	With RDs		Without RDs	
	Male	Female	Male	Female
3	8	5	8	7
4	10	5	13	16
Total	18	10	21	23

**Table 2 brainsci-14-00428-t002:** Inference frequencies during reading.

Genre	Inference Type	With RDs		Without RDs		Statistic (*p*-Value)
		Mean	SE	Mean	SE	
Expository	Correct	0.23	0.05	0.66	0.05	183.5 (*p* < 0.001)
	Incorrect	0.17	0.04	0.08	0.02	753.0 (*p* > 0.05)
	No	0.60	0.08	0.26	0.05	915.0 (*p* < 0.001)
Narrative	Correct	0.29	0.05	0.70	0.04	157.5 (*p* < 0.001)
	Incorrect	0.12	0.03	0.06	0.01	714.0 (*p* > 0.05)
	No	0.59	0.07	0.24	0.04	949.5 (*p* < 0.001)

**Table 3 brainsci-14-00428-t003:** Inference frequencies during early and late reading of the expository text.

Group	Inference Type	Early		Late		Statistic (*p*-Value)
		Mean	SE	Mean	SE	
RD	Correct	0.24	0.04	0.22	0.04	29.0 (*p* > 0.05)
	Incorrect	0.15	0.04	0.20	0.04	17.0 (*p* > 0.05)
	No	0.61	0.07	0.58	0.08	23.0 (*p* > 0.05)
Without RD	Correct	0.65	0.05	0.67	0.06	159.0 (*p* > 0.05)
	Incorrect	0.09	0.03	0.07	0.03	58.0 (*p* > 0.05)
	No	0.26	0.05	0.26	0.05	99.0 (*p* > 0.05)

**Table 4 brainsci-14-00428-t004:** Inference frequencies during early and late reading of the narrative text.

Group	Inference Type	Early		Late		Statistic (*p*-Value)
		Mean	SE	Mean	SE	
RD	Correct	0.24	0.06	0.33	0.06	74.5 (*p* > 0.05)
	Incorrect	0.07	0.03	0.15	0.04	13.0 (*p* > 0.05)
	No	0.69	0.07	0.52	0.07	24.0 (*p* < 0.01)
Without RD	Correct inference	0.60	0.05	0. 79	0.04	144.0 (*p* < 0.001)
	Incorrect inference	0.05	0.02	0.06	0.02	68.0 (*p* > 0.05)
	No inference	0.35	0.05	0.16	0.04	55.0 (*p* < 0.001)

## Data Availability

Available upon request. The data are not publicly available due to Personal Information Protection Act.

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
