# Peer review of "On the Dynamics of Inferential Behavior while Reading Expository and Narrative Texts"

_brainsci, 2024, doi:10.3390/brainsci14050428_

Round 1
Reviewer 1 Report
Comments and Suggestions for Authors
Dear Authors,
The focus of the manuscript is the current issue of researching the processes involved in comprehending written discourse across various genres. Students with reading difficulties and normative readers are compared. Still, there are a few issues with the manuscript that need to be fixed.
1. In the introduction, literature should be added on the comparative linguistic characteristics of expository and narrative texts that influence reading strategies.
2. In line 66, the authors discuss the relevance of taking into account the influence of cognitive resources on the dynamics of text comprehension, but such methods were not used in the current study. In that case, there is no point in discussing this issue.
3. It is necessary to specify the criteria for the selection of subjects into each group according to RA-RCP data.
4. In the methodology section, it is necessary to give the linguo-statistical characteristics of the texts: average sentence length, min, max. Characterize the content of the texts and their macrostructure. Indicate to which area of knowledge the texts belong and how this correlates with pupils' knowledge.
5. It is necessary to indicate how the text was presented: as a whole or by sentences, parameters of stimuli (font, size, line length), and exposure time. Randomization of the order of presentation of texts.
6. Explain the methodology used to determine the subjects' overall comprehension level for each of the texts.
7. Provide information about the number and quality of the coders' professional experience.
8. In lines 109–121, you describe the procedure for classifying responses into three types. Illustrations of each type should be given. Indicate how the first five and the last five sentences relate in terms of information load and linguistic characteristics. How was the frequency index calculated?
10. The results should include a table of descriptive statistics for the groups for all measured indicators, by age and gender.
11. It is necessary to provide data on the assessment of the general level of text comprehension in both groups and the reliability of intergroup differences.
12. The text in lines 158–162 should be moved to the discussion section.
13. The statement given in lines 169–171 seems questionable. This difference may be caused by the difference in the content of the sentences and the information load of the first five and the next five sentences.
Reviewer 2 Report
Comments and Suggestions for Authors
I would like to refer to the article you have prepared which is a very good basis for the topic you are working on. These are the points where I think they need improvement and strengthening:
1. In 1.1. the theoretical framework needs to be strengthened as modern studies on the genre of the text are mentioned in order to better introduce the reader to the subject.
In 1.2 part of your reasoning is based on a hypothesis, it would be good to document and support with literature this issue and your research questions better connect to the research gap, in this form the way they are stated seems like not connects naturally to the text and the results of existing knowledge.
2. In terms of your methodology it is important to mention whether a similar protocol has been used as to the material, the administration procedure, its assessors as well as in 2.3 has the assessor experience factor been taken into account? It is good to describe the part that existed to ensure the objectivity and reliability of the measurements.
In 2.3, the 3 criteria that you took into account are generally not immediately perceptible, especially the first one is divided into intermediate sub-stages and confuses with the wording of the second factor. Please clarify and document each factor.
In 2.4, does your sample follow a normal distribution and was the t-test performed? It would be good if it were analyzed and presented more fully.
Caution throughout the text: Do not use the term normal participants, prefer the control group and the experimental group.
Τhe discussion needs strengthening, and each finding should be linked to existing knowledge and literature to substantiate your results and guide the reader.
What are the clinical implications at the end of your study, must be answered more specifically.
Be sure to highlight limitations and future extensions of the research.
Reviewer 3 Report
Comments and Suggestions for Authors
This study examines changes in inferential behavior when reading different genres. The inferential behavior of 28 students with reading disabilities (RD) and 44 students without RD was quantified during reading expository and narrative texts.
The study can be interesting, since it can shed light on improvement and reading comprehension.
However, it is necessary to make it a little more rigorous. It must indicate how the measurements were obtained, and it must also indicate the results, with means, and estimation of the effect in tables, not in figures.
The introduction should make it clear how the inference is identified in each subtype, that is, what references there are to it.
And in the methodology, identify the experts who evaluate, and based on what criteria they identify the inference, since it is the basis of the article.
On the other hand, the images, I don't know if they should appear, but without tables, the text is not understandable.
Comments on the Quality of English Language This study examines changes in inferential behavior when reading different genres. The inferential behavior of 28 students with reading disabilities (RD) and 44 students without RD was quantified during reading expository and narrative texts. The study can be interesting, since it can shed light on improvement and reading comprehension. However, it is necessary to make it a little more rigorous. It must indicate how the measurements were obtained, and it must also indicate the results, with means, and estimation of the effect in tables, not in figures. The introduction should make it clear how the inference is identified in each subtype, that is, what references there are to it. And in the methodology, identify the experts who evaluate, and based on what criteria they identify the inference, since it is the basis of the article. On the other hand, the images, I don't know if they should appear, but without tables, the text is not understandable.Author Response
Please see the attachment.

Round 2
Reviewer 1 Report
Comments and Suggestions for Authors
In this version, I have no comment

Reviewer 2 Report
Comments and Suggestions for Authors
All my suggestions have been addressed and the text has been significantly upgraded.